# Gas Permeation Model of Mixed-Matrix Membranes with Embedded Impermeable Cuboid Nanoparticles

**DOI:** 10.3390/membranes10120422

**Published:** 2020-12-15

**Authors:** Haoyu Wu, Maryam Zamanian, Boguslaw Kruczek, Jules Thibault

**Affiliations:** 1Department of Chemical and Biological Engineering, University of Ottawa, 161 Louis Pasteur St., Ottawa, ON K1N 6N5, Canada; hwu055@uottawa.ca (H.W.); bkruczek@uottawa.ca (B.K.); 2Department of Biosystems Engineering, Ferdowsi University of Mashhad, Azadi Square, Mashhad, Razavi Khorasan Province 9177948974, Iran; mzama074@uottawa.ca

**Keywords:** mixed-matrix membranes, impermeable nanoparticles, three-dimensional modelling, relative permeability, predictive permeability model

## Abstract

In the packaging industry, the barrier property of packaging materials is of paramount importance. The enhancement of barrier properties of materials can be achieved by adding impermeable nanoparticles into thin polymeric films, known as mixed-matrix membranes (MMMs). Three-dimensional numerical simulations were performed to study the barrier property of these MMMs and to estimate the effective membrane gas permeability. Results show that horizontally-aligned thin cuboid nanoparticles offer far superior barrier properties than spherical nanoparticles for an identical solid volume fraction. Maxwell’s model predicts very well the relative permeability of spherical and cubic nanoparticles over a wide range of the solid volume fraction. However, Maxwell’s model shows an increasingly poor prediction of the relative permeability of MMM as the aspect ratio of cuboid nanoparticles tends to zero or infinity. An artificial neural network (ANN) model was developed successfully to predict the relative permeability of MMMs as a function of the relative thickness and the relative projected area of the embedded nanoparticles. However, since an ANN model does not provide an explicit form of the relation of the relative permeability with the physical characteristics of the MMM, a new model based on multivariable regression analysis is introduced to represent the relative permeability in a MMM with impermeable cuboid nanoparticles. The new model possesses a simple explicit form and can predict, very well, the relative permeability over an extensive range of the solid volume fraction and aspect ratio, compared with many existing models.

## 1. Introduction

### 1.1. Mixed-Matrix Membranes (MMMs) as Barrier Materials

Barrier thin films are widely used in food and beverage packaging, coating, drug release, fuel cells, and batteries [1,2,3]. Many of these films are made of polymer materials, of which the barrier properties are constrained due to the limitation in their thicknesses. By adding impermeable inorganic fillers into the polymeric matrix of the thin films, the permeation of water and gas molecules in barrier materials is further prevented or delayed. The embedding of a homogenously-dispersed inorganic filler into the continuous phase of a polymeric matrix is typically referred to as a mixed-matrix membrane (MMM). The shape, volume fraction, and dimensions of the nanoparticles within the polymer are essential factors that determine the effective gas permeability of an MMM. Wolf et al. [3] performed an exhaustive review of the effects of the shape of the fillers on the barrier properties of polymer/impermeable nanocomposite membranes. Based on 1000 published data, their results showed a very wide variation of the relative permeability of MMMs. They nevertheless confirmed the superiority of layered nanoparticles to decrease the membrane relative permeability compared to isodimensional and elongated nanoparticles. Many researchers [4,5,6] also believe that fillers have high aspect ratios create a more tortuous path for diffusion and therefore improves the barrier properties by orders of magnitude even for small volume fractions of the fillers [6]. For this reason, flake-like or plate-like fillers such as mica and clay minerals (hectorite, saponite, and montomorillonite) are studied intensively due to their high aspect ratios [6,7,8]. The aspect ratio is commonly defined as the ratio of the mean diameter of a circle of the same area of the filler to the mean thickness of the filler. There is a clear need to better understand the effect of the shape of nanoparticles on the relative permeability of MMMs and to have reliable theoretical predictions.

A number of researchers have studied and proposed models to characterize the permeation for both ideal and non-ideal morphology of MMMs [9]. Ideal membrane morphology refers to a two-phase membrane system with good adherence between the continuous phase and the discontinuous phase. The majority of theoretical models to predict the effective permeability of MMMs were developed based on ideal membrane morphology, such as the models proposed by Maxwell [10], Bruggeman [11], Lewis–Nielsen [12,13], Pal [14], Cussler [15,16], and Bharadwajl [17]. Non-ideal morphology refers to MMMs for which a non-ideal polymer-particle interface exists due to poor polymer−particle adhesion, polymer-packing disruption near the dispersed particles, and repulsive forces between the two phases. MMMs are sometimes characterized using a three-phase system, having the interface voids as the third phase, as proposed in a few modified models [18]. However, in reality, the interfacial defects are the result of many factors and vary from experiment to experiment such that the accurate determination of the exact interface morphology is a challenge. In this study, we assume that the dispersed phase is nonporous (impermeable) and ideal MMM morphology exists with no defects and no distortion at the filler−polymer interface. It is also assumed that, for a given MMM, the fillers have identical shapes and sizes, and they are uniformly distributed within the polymeric membrane. The case where interfacial defects exist is briefly discussed at the end of this study.

### 1.2. Main Prediction Models of the Relative Permeability of MMMs

Many models, both analytical and numerical, for predicting the relative permeability of MMMs have been proposed over the years. Six of these analytical predictive models, considered most pertinent in this investigation, are presented in Table 1. These analytical models were proposed by Maxwell [10], Bruggeman [11], Lewis–Nielsen [12,13], Pal [14], Cussler [15,16], and Bharadwajl [17]. These models have been developed to describe the permeation of a species through MMMs and to estimate their relative permeability (*P_r_*):(1)$$P_{r}=\frac{P_{eff}}{P_{c}},$$
where *P_eff_* is the effective permeability of the MMM and *P_c_* is the permeability of the continuous phase, i.e., that of a neat polymeric membrane. In all these models, *P_r_* depends on the filler volume fraction ϕ, the permeability of the continuous phase *P_c_*, and the permeability of the dispersed phase *P_d_*. For a MMM with impermeable particles, *P_d_* = 0, the mathematical expressions of Table 1 are significantly reduced. The first model used to predict the permeation properties of MMM was the Maxwell’s model, a model initially proposed to estimate the dielectric properties of polymer composites. Among all models listed, the Maxwell’s model is the most commonly used. Besides the permeability coefficients of the continuous phase (the polymer) and the dispersed phase (the fillers), the Maxwell’s model uses only the volume fraction ϕ as a model parameter, regardless of the particle shape, size distribution, and particle dispersion. The model is only applicable to a dilute two-phase MMM system containing spherical particles with ϕ < 0.2. Bruggeman [11], also attempting to predict the dielectric properties of composite materials with randomly dispersed particles, modified the Maxwell’s model to predict the relative permeability over a larger range of the volume fraction. Lewis and Nielsen [12,13] studied the mechanical properties of composite materials by experimentally exploring the relationship between the relative elastic modulus of composite materials and the volume fraction of its spherical fillers. The equation introduced an additional function *ψ* taking into account the maximum packing fraction $\phi_{m}$. The model was often applied to predicting the *P_r_* of MMMs. Pal’s model was originally developed to determine the effective thermal conductivity of composite materials using the differential effective medium approach. This model also introduced an additional parameter, the maximum packing fraction $\phi_{m}$. The Bruggeman’s model is a special case of Pal’s model when $\phi_{m}$ = 1.0. Most models in Table 1 idealized the filler geometry and investigated only the effect of the filler volume fraction ϕ. To understand the impact of the filler’s geometry, models such as the one proposed by Cussler’s group relate the relative permeability to an aspect ratio and the volume fraction ϕ [15,16]. They extended the Maxwell’s model by studying regular and random arrays of high aspect ratio impermeable particles, such as flakes and lamellae. They finally verified their model by performing experiments with the measurement of electrical resistance of salt solutions. Another well-known model is the two-dimensional model proposed by Bharadwajl [17]. Bharadwaj modified Nielsen’s model and performed a theoretical study on the effects of filler sheet length, concentration, orientation, and degree of delamination on the relative permeability of MMMs. He concluded that dispersing a long sheet of inorganic filler in a polymer matrix was particularly beneficial for barrier properties.

As previously stated, based on the Wolf et al. [3] review, layered impermeable nanoparticles are more effective in reducing the permeability of MMMs than their isodimensional and elongated counterparts. Yet most of the available analytical models, including the majority of those listed in Table 1, are based on spherical nanoparticles. Wolf et al. [3] clearly demonstrated that the relative permeability of MMMs with impermeable fillers as a function of the volume fraction often behaves differently than expected, which gives rise to a series of hypotheses to explain the different trends observed. To be able properly determine the reasons for non-ideal behavior, it is essential to compare the relative permeability obtained experimentally with the one expected for an ideal MMM, thereby serving as a reference relative permeability. Therefore, the objective of this paper is two-fold. First, we would like to show that the models listed in Table 1 cannot be used to accurately predict *P_r_* of MMMs with dispersed layered nanoparticles over a wide range of volume fractions. Secondly, to develop a simple, yet general, mathematical model to predict *P_r_* of MMMs for different geometries and dimensions of the filler particles in a 3D setting. Considering that a general and easily identifiable analytical model, which covers all geometries of nanoparticles, may or may not exist, as an intermediate step between the numerical solution and an analytical model to predict *P_r_*, we also demonstrate the applicability of an artificial neural network model for the prediction of *P_r_*. 

## 2. Gas Transport in an MMM

A schematic diagram of a MMM is shown in Figure 1. It is assumed that the nanoparticles are uniformly distributed throughout the whole membrane, such that it is possible to define an elementary cubic unit comprised of a single nanoparticle located at the center of the polymer cube. The membrane, therefore, consists of multiple identical elementary units juxtaposed to form the complete membrane. It can be easily shown that the permeability of a permeating species is identical for both a single elementary unit and the whole membrane [20,21]. Consequently, to estimate the permeability of the entire MMM membrane, it is sufficient to solve Fick’s second law of diffusion numerically for a single elementary unit. In this investigation, the polymer phase was assumed to be isotropic, and the nanoparticle was located at the center of an elementary unit was assumed impermeable to the permeating gas. Two different types of impermeable nanoparticles often suggested as fillers in barrier films were used in this investigation. These were spherical and cuboid nanoparticles. An example of the former is TiO_2_ nanoparticles, and an example of the latter is montmorillonite (MMT) clay nanoparticles. The dimensions of each cubic elementary unit were *L_x_*, *L_y_*, and *L_z_* (Figure 1), whereas the spherical nanoparticle was defined with its diameter *d_p_*, and the cuboid nanoparticle dimensions were *x_p_*, *y_p_*, and *z_p_*.

Gas permeation across a membrane follows three steps: sorption, diffusion, and desorption. The sorption and desorption processes follow Henry’s law. For a membrane of constant solubility *S*, the concentration of the migrating species at each surface of the membrane is proportional to the surrounding pressure as described in Equation (13). It was assumed that the concentrations of gas molecules in the gas phase and within the membrane at the two gas-membrane interfaces were in instantaneous equilibrium. The diffusion process follows the Fickian’s first and second laws, as expressed in Equations (14) and (15). In a neat polymeric membrane, the diffusion occurs solely in the direction (*y*) perpendicular to the membrane. When permeable or impermeable particles were embedded within the polymeric matrix of the membrane, the diffusion occurred in the *x*, *y*, and *z* directions in Cartesian coordinates.
(13)C=S p
(14)$$\vec{J}={\vec{J}}_{x}+{\vec{J}}_{y}+{\vec{J}}_{z}=-D\left(\frac{\partial C}{\partial x}\vec{i}+\frac{\partial C}{\partial y}\vec{j}+\frac{\partial C}{\partial z}\vec{k}\right)$$
(15)$$\frac{\partial C}{\partial t}=D\left[\frac{\partial^{2}C}{\partial x^{2}}+\frac{\partial^{2}C}{\partial y^{2}}+\frac{\partial^{2}C}{\partial z^{2}}\right]$$
where *p* is the gas pressure adjacent to a membrane surface, and *C* is the concentration of the migrating species within the MMM, $\vec{J}$ is the time-varying local permeation flux (a vector), *t* is the time, and *x*, *y*, and *z* are the Cartesian coordinates to account for the three-dimensional diffusion within the membrane where *y* is the main direction of permeation. In this investigation, it was assumed that the membrane (continuous phase) had a constant and isotropic diffusivity *D* whereas the nanoparticles (dispersed phase) was impermeable and had a nil diffusivity. It was further assumed that the diffusivity was independent of concentration. In this investigation, we were, therefore, considering an ideal MMM with impermeable nanoparticles.

## 3. Methodology

The three-dimensional form of the Fick’s second law of diffusion (Equation (15)) can be discretized using finite differences. Equation (16) represents the explicit discretization of Equation (15) for an interior mesh point (*i*, *j*, *k*) that allows calculating the concentration at the next time step knowing the current concentration at a given mesh point within the membrane. The change of concentration at mesh point (*i*, *j*, *k*), as a function of time, depends on the current concentration at point (*i*, *j*, *k*) and the concentrations at the six neighboring mesh points, as illustrated in Figure 2.
(16)$$C_{i,j,k}^{m+1}=C_{i,j,k}^{m}+\Delta t\left\{\begin{array}{l} \frac{2D_{c}}{\Delta x_{i+1}+\Delta x_{i}}\left[\frac{C_{i+1,j,k}^{m}-C_{i,j,k}^{m}}{\Delta x_{i+1}}-\frac{C_{i,j,k}^{m}-C_{i-1,j,k}^{m}}{\Delta x_{i}}\right]\\ +\frac{2D_{c}}{\Delta y_{j+1}+\Delta y_{j}}\left[\frac{C_{i,j+1,k}^{m}-C_{i,j,k}^{m}}{\Delta y_{j+1}}-\frac{C_{i,j,k}^{m}-C_{i,j-1,k}^{m}}{\Delta y_{j}}\right]\\ +\frac{2D_{c}}{\Delta z_{k+1}+\Delta z_{k}}\left[\frac{C_{i,j,k+1}^{m}-C_{i,j,k}^{m}}{\Delta z_{k+1}}-\frac{C_{i,j,k}^{m}-C_{i,j,k-1}^{m}}{\Delta z_{k}}\right] \end{array}\right\}$$

It is important to reiterate that it is only necessary to solve for a single elementary unit to estimate the effective permeability of the MMM. The elementary unit consists of a polymer cuboid (*L_x_* by *L_y_* by *L_z_*) with a single nanoparticle at its center. To solve numerically for the temporal variation of the concentration at all points within the membrane, it was necessary to define the initial and boundary conditions. Concerning the initial condition, i.e., before the concentration step-change on one side of the membrane at time *t* = 0, a nil concentration was assumed throughout the membrane (Equation (17)). Concerning the boundary conditions, twelve relations were required to define the problem: Six boundary conditions at the periphery of the polymeric elementary unit and six boundary conditions at the polymer-solid interfaces. Equation (18) provided the boundary conditions on both sides of the membrane (*y*-axis was the permeation direction). It was assumed that at the onset of permeation, a step-change in the gas pressure was applied to the upstream side of the membrane (*y* = 0) whereas the gas pressure in the downstream side of the membrane (*y* = *L_y_*) was kept under perfect vacuum. The resulting concentrations in the membrane at both surfaces were simply the product of the neighboring pressure and the solubility S. Because all elementary units were identical, symmetry conditions prevailed at the other four faces of the polymer parallelepiped as expressed in Equation (19) for BC_3–6_. Since the nanoparticle in the center of the elementary unit was impermeable, the mass flux at each of the six faces of the nanoparticle, assuming a cuboid nanoparticle, was zero (see BC_7–12_ in Equation (20)). The latter boundary conditions imply that the concentration of the migrating species inside the nanoparticle was zero.
(17)$$\begin{array}{cc} \mathrm{IC}: & C_{i,j,k}^{t\;<\;0}=0 \end{array}\;\forall i,j,k$$
(18)$$\begin{array}{ll} {\mathrm{BC}}_{1}:\;C_{i,y\;=\;0,k}=p_{0}S & \;\forall i,k\\ {\mathrm{BC}}_{2}:\;C_{i,y\;=\;L_{y},k}=0 & \;\forall i,k \end{array}$$
(19)$${\mathrm{BC}}_{3–6}:\;{\left.\frac{\partial C}{\partial x}\right|}_{x\;=\;0}={\left.\frac{\partial C}{\partial x}\right|}_{x\;=\;L_{x}}={\left.\frac{\partial C}{\partial z}\right|}_{z\;=\;0}={\left.\frac{\partial C}{\partial z}\right|}_{z\;=\;L_{z}}=\;0$$
(20)$$\begin{array}{cc} {\mathrm{BC}}_{7–8}:\;{\left.\frac{\partial C}{\partial x}\right|}_{x\;=\;\frac{\left(L_{x}\;-\;x_{p}\right)}{2}}=\;{\left.\frac{\partial C}{\partial x}\right|}_{x\;=\;\frac{\left(L_{x}\;+\;x_{p}\right)}{2}}=0 & \;\forall \left[\left(\frac{\left(L_{y}\;-\;y_{p}\right)}{2}\leq y\leq \frac{\left(L_{y}\;+\;y_{p}\right)}{2}\right)\cap \left(\frac{\left(L_{z}\;-\;z_{p}\right)}{2}\leq z\leq \frac{\left(L_{z}\;+\;z_{p}\right)}{2}\right)\right]\\ {\mathrm{BC}}_{9–10}:\;{\left.\frac{\partial C}{\partial y}\right|}_{y\;=\;\frac{\left(L_{y}\;-\;y_{p}\right)}{2}}=\;{\left.\frac{\partial C}{\partial y}\right|}_{y\;=\;\frac{\left(L_{y}\;+\;y_{p}\right)}{2}}=0 & \;\forall \left[\left(\frac{\left(L_{x}\;-\;x_{p}\right)}{2}\leq x\leq \frac{\left(L_{x}\;+\;x_{p}\right)}{2}\right)\cap \left(\frac{\left(L_{z}\;-\;z_{p}\right)}{2}\leq z\leq \frac{\left(L_{z}\;+\;z_{p}\right)}{2}\right)\right]\\ {\mathrm{BC}}_{11–12}:\;{\left.\frac{\partial C}{\partial z}\right|}_{z\;=\;\frac{\left(L_{z}\;-\;z_{p}\right)}{2}}=\;{\left.\frac{\partial C}{\partial z}\right|}_{z\;=\;\frac{\left(L_{z}\;+\;z_{p}\right)}{2}}=0 & \;\forall \left[\left(\frac{\left(L_{x}\;-\;x_{p}\right)}{2}\leq x\leq \frac{\left(L_{x}\;+\;x_{p}\right)}{2}\right)\cap \left(\frac{\left(L_{y}\;-\;y_{p}\right)}{2}\leq y\leq \frac{\left(L_{y}\;+\;y_{p}\right)}{2}\right)\right] \end{array}$$

The upstream and downstream fluxes were obtained from the concentration gradients at the upstream and downstream gas-membrane interfaces using Equations (21) and (22). Convergence to the steady permeation rate was assumed to be reached when the fluxes at the upstream and downstream surfaces were the same, within 0.001%. The effective permeability (*P_eff_*) was calculated from the steady flux, the thickness *L_y_* of the elementary unit and the feed pressure difference between the two sides of the elementary unit of the membrane (Δ*p*), as expressed in Equation (23).
(21)$$J_{y=0}=-D_{c}\frac{\partial C_{y=0}}{\partial y}$$
(22)$$J_{y=L_{y}}=-D_{c}\frac{\partial C_{y=L_{y}}}{\partial y}$$
(23)$$P_{eff}=\frac{J_{y}L_{y}}{\frac{\left(p_{0}-0\right)}{RT}}$$

In Equations (21) and (22), *J_y_*_=0_ is the upstream permeation flux and *J_y=Ly_* is the downstream permeation flux.

## 4. Results and Discussion

### 4.1. Numerical Simulation Results

This section presents the results for the series of 3D simulated permeation experiments of gas molecules across a polymeric elementary unit with an impermeable nanoparticle at its center. The impermeable nanoparticle at the center of the elementary unit was either a sphere or a cuboid, representing the shape of the TiO_2_ and the MMT nanoparticles, respectively. For a spherical nanoparticle, the diameter was varied whereas, for a cuboid nanoparticle, the relative thickness, *y_p_/L_y_*, and the aspect ratio *q* were varied in order to assess the effect of the size and the shape of nanoparticles on the relative permeability (*P_r_*) of MMMs. Similar to the Cussler’s definition but in a 3D setting, the aspect ratio (*q*) is defined in a dimensionless form using Equation (24):(24)$$q=\frac{\sqrt{x_{p}z_{p}}}{y_{p}}$$

In the current investigation, we assumed that the *x-z* plane of a cuboid nanoparticle was parallel to the surface of the membrane, i.e., *q* was the square root of the projected area divided by the thickness of the nanoparticle.

Figure 3 presents the relative permeability of MMMs obtained from the simulated permeation experiments with spherical nanoparticles for a filler volume fraction ϕ ranging from 0 to 0.52 and with cuboid nanoparticles for a ϕ ranging from 0 to 0.77. Since *q* has a significant impact on *P_r_*, the results for the cuboid nanoparticles were grouped over a series of narrow ranges of the aspect ratio: from 0.51–0.56 to 7.00–7.29. Many more numerical results were obtained than those presented in Figure 3, and they will all be used in developing a model to predict *P_r_*. When *q* is large, the cuboid nanoparticle is a thin flat sheet with a large projected area. The migrating species must circumvent to reach the permeate side of the membrane, thereby reducing the permeability of the membrane. On the other hand, when *q* is small, the cuboid has a small projected area with a large thickness (*y_p_*). 

As expected, the relative permeability decreases with an increase in the volume fraction occupied by the nanoparticles within each q. It is evident that the aspect ratio has a major impact on the effective permeability of the membrane. Results of Figure 3 show that *P_r_* as a function of ϕ for MMMs with an impermeable sphere and an impermeable cube (having an aspect ratio of unity) was nearly identical. For impermeable cuboid nanoparticles with *q* smaller than unity, the relative permeability is greater than the relative permeability of MMMs with spherical and cubic nanoparticles. In contrast, when *q* is larger than unity, the situation is exactly the opposite. For a membrane with cuboid nanoparticles with the same ϕ, the greater the value of *q*, the smaller the value of *P_r_*. For all results presented in this investigation, the permeability of the continuous phase was assumed to equal to 5 × 10^−12^ m^2^/s (5.0 × 10^−11^ m^2^/s diffusivity and 0.10 solubility). However, it is important to note that the relative permeability was independent of the permeability of the continuous polymeric phase since the filler was impermeable. A different polymer permeability would only affect the time required to reach steady-state permeation and the steady-state permeation flux.

With the numerous estimations of *P_r_* obtained numerically in this investigation, it was desired to verify if any correlation available in the literature could adequately predict the relative permeability of cuboids with large values of the *q*. Figure 4 compares the relative permeability obtained numerically with the one obtained using the six correlations of Table 1 for spherical nanoparticles for a value of ϕ in the range [0.00, 0.52], and cuboid nanoparticles for four different narrow interval ranges of *q*: [2.0, 2.3], [3.0, 3.2], [5.0, 5.2], and [7.0, 7.3]. Results confirm that the relative permeability of spherical nanoparticles predicted by the model proposed by Maxwell is nearly identical to the one obtained numerically. The predicted *P_r_* by the Maxwell model starts to deviate for a volume fraction in the vicinity of 0.25. On the other hand, three models (Bruggeman (BGM), Lewis–Nielsen (LN), and Pal (PAL)) accurately predict *P_r_* over the whole range of ϕ. Only the models proposed by Cussler [15,16] and Bharadwajl [17], which were not developed for spheres, show more significant deviations. Based on the results of Figure 3, it is not surprising that the same four models were equally able to predict the relative permeability of the cuboid nanoparticle with an aspect ratio of unity (results not shown). For the prediction of *P_r_* for cuboid nanoparticles with a *q* value other than unity, Cussler’s model predicts better than the four models for the two larger *q* values [5.0, 5.2] and [7.0, 7.3]; however, the deviations are still significant. The model proposed by Bharadwajl [17], which includes some geometrical parameters, can reasonably represent the data for small ϕ. However, this model becomes less accurate in predicting *P_r_* for higher ϕ. All the other models over-predicted *P_r_* of MMMs with cuboid nanoparticles when *q* was greater than unity and under-predicted *P_r_* when *q* was smaller than unity. It is evident that the available analytical models fail to predict accurately the relative permeability of MMMs with dispersed impermeable nano-cuboids with *q* > 1, in particular when ϕ becomes larger. The greater the value of *q*, the more significant the deviation between the simulated and predicted relative permeability is even at very small ϕ. For example, for *q* = 5, the deviation between the simulated *P_r_* and the one predicted by the best existing model (BWD) becomes evident at ϕ ~ 0.02. 

These results show that it is imperative to develop a model that would predict the permeation behavior of MMMs embedding impermeable nano-cuboids with *q* different from unity.

### 4.2. Artificial Neural Network Model for the Prediction of the Relative Permeability

If the purpose is to find rapidly an accurate model that could be used for predicting the dependent variable, such as the relative permeability (*P_r_*), an artificial neural network can be used. Artificial neural networks are now commonly used for a myriad of engineering applications. The high degree of plasticity of its structure is the main reason for its ability to efficiently represent the underlying causal relationship between input and output data. In this investigation, a three-layer feedforward neural network (FFNN) was used to predict *P_r_* as a function of some input variables. Cybenko [22] showed that a three-layer FFNN was sufficient to encapsulate any input-output relationship if a sufficient number of neurons are used.

In this investigation, the FFNN consisted of an input layer, a hidden layer, and an output layer, as shown in Figure 5. The input layer contains many neurons corresponding to the number of independent variables plus the bias neuron. The input layer transfers each set of independent variables to each functional neuron of the hidden layer. Each functional neuron of the hidden layer performs a weighted summation of all process inputs and a nonlinear transformation of the weighted summation to generate the output of each neuron of the hidden layer. The outputs of the hidden layer, including the bias neuron of the hidden layer, are then sent to the output neuron. The output neuron performs the same task as the neurons of the second layer to generate the final output of the FFNN. A sigmoid function (Equation (25)) was used as the transfer function for the hidden and output neurons. In this investigation, the inputs and outputs of the FFNN of Figure 5 are, by definition, already scaled between 0 and 1, so that it was not necessary to scale them as it is usually done.
(25)$$f(\Psi )\;=\;\frac{1}{1\;+\;e^{-\Psi }}$$

Various input data vectors were explored in order to find the simplest neural network structure to accurately predict *P_r_*. The two simplest structures of the FFNN were the ones that used only the relative dimensions of the nanoparticles within an elementary unit. In one case, the three relative lengths, *x_p_/L_x_*, *y_p_/L_y_* and *z_p_/L_z_*, were used. In the other case, the relative projected area (*x_p_z_p_/L_x_L_z_*) and the relative thickness (*y_p_/L_y_*) as shown in Figure 5, were used. To determine the neural model for the prediction of *P_r_*, 359 data points obtained by solving numerically the governing partial differential equation were divided equally into a training and a validation data set. The quasi-Newton nonlinear regression algorithm was used to adjust the weights of the FFNN that minimize the sum of squares of the training data set. At each iteration, the sum of squares of the validation data set was also evaluated, and the set of weights that minimize the sum of squares of the validation data set was retained. The coefficient of regression for the FFNN with six hidden neurons (including the bias) was 0.9998 for both neural network structures mentioned above. The parity plot based on the FFNN of Figure 5 is presented in Figure 6.

The FFNN was able to predict very accurately *P_r_* of MMMs containing impermeable cuboid nanoparticles. The accuracy of the neural model was excellent over the entire range of *x_p_z_p_/L_x_L_z_* and *y_p_/L_y_* as the parity plot of Figure 6 shows. The most significant deviation was observed for thin cuboids covering nearly the entire *x-z* area of the elementary unit, thereby associated with low relative permeability. The latter condition was, however, extreme and will not be encountered in reality. The thinner the cuboid, the larger the deviation. It is believed that the FFNN can be used with confidence to predict *P_r_*. The mathematical model of the FFNN is given in Equations (26)–(28). The results obtained with the neural network suggest that a strong relationship exists between the relative projected area and the relative thickness. It is, therefore, hopeful of developing an analytical model with the two geometrical parameters. It differs from the traditional modelling approach mostly based on the volume fraction ϕ [23].
(26)$$\begin{array}{c} {\;W}_{ij}\;\\ \left|\begin{array}{c} H_{1}\\ H_{2}\\ H_{3}\\ H_{4}\\ H_{5} \end{array}\right|\;=\;f\left(\left|\begin{array}{ccc} -12.13 & 0.7200 & -4.087\\ -4.774 & -0.8497 & 0.9264\\ 47.18 & 0.4026 & -50.00\\ -3.616 & -0.1625 & 4.961\\ 1.441 & -3.201 & -3.831 \end{array}\right|•\left|\begin{array}{c} \frac{x_{p}z_{p}}{L_{x}L_{z}}\\ \frac{y_{p}}{L_{y}}\\ 1 \end{array}\right|\right)\\ {\;W\;\prime }_{jk}\\ {\hat{P}}_{r}\;=\;f\left(\left|\begin{array}{cccccc} 44.95 & 3.392 & -49.94 & 14.15 & 25.77 & -13.92 \end{array}\right|•\left|\begin{array}{c} H_{1}\\ H_{2}\\ H_{3}\\ H_{4}\\ H_{5}\\ 1 \end{array}\right|\right) \end{array}$$
(27)$$H_{j}\;=\;\frac{1}{1\;+\;e^{-(W_{1j}\frac{x_{p}z_{p}}{L_{x}L_{z}}\;+\;W_{2j}\frac{y_{p}}{L_{y}}\;+\;W_{3j})}}\;=\;\frac{1}{1\;+\;Exp\left(-\sum_{i=1}^{3}W_{ij}\;X_{i}\right)}$$
(28)$${\hat{P}}_{r}\;=\;\frac{1}{1\;+\;e^{-({W^{\prime }}_{11}H_{1}\;+\;{W^{\prime }}_{21}H_{2}\;+\;{W^{\prime }}_{31}H_{3}\;+\;{W^{\prime }}_{41}H_{4}\;+\;{W^{\prime }}_{51}H_{5}\;+\;{W^{\prime }}_{61})}}\;=\;\frac{1}{1\;+\;Exp\left(-\sum_{j=1}^{J}{W^{\prime }}_{j1}\;H_{j}\right)}$$

### 4.3. New Analytical Model for Pr of MMMs with Impermeable Cuboid Nanoparticles

Although the neural network model presented in the previous section provides excellent predictions of the relative permeability of cuboid nanoparticles for a wide range of geometrical parameters, it does not provide a clear insight into the impact of the input variables on *P_r_* as the analytical models listed in Table 1 do. On the other hand, these models, except for the Bharadwajl’s correlation [17], fail to accurately predict *P_r_* for cuboid nanoparticles with *q* different from unity. Therefore, it is desired to propose a new analytical model that will be valid for the widest possible ranges of ϕ and the geometrical parameters of the cuboid nanoparticles.

To clearly elucidate the relationship between the shape and the relative dimensions of nanoparticles and *P_r_*, multivariate covariance analysis was performed to assess the underlying correlation between all possible geometrical factors and *P_r_*. For this analysis, the Pearson correlation coefficient (PCC), defined in Equations (29) to (31), was used [24].
(29)$$\sigma =\sqrt{\frac{\sum_{i=1}^{n}{\left(E_{i}-\bar{E}\right)}^{2}}{n-1}}$$
(30)$$\mathrm{Cov}\left(E_{A}{,E}_{B}\right)=\frac{\sum_{i=1}^{n}\left(E_{\mathrm{Ai}}-E_{A}\right)\left(E_{\mathrm{Bi}}-E_{B}\right)}{n-1}$$
(31)$$\mathrm{PCC}\left(E_{A},E_{B}\right)=\frac{\mathrm{cov}\left(E_{A},E_{B}\right)}{\sigma_{A}\sigma_{B}}$$
where *σ* is the standard deviation of the variable E, Cov(E_A_, E_B_) is the covariance of E_A_ and E_B_, E_A_ and E_B_ are the average values of E_A_ and E_B_, and PCC(E_A_, E_B_) is the Pearson correlation coefficient between E_A_ and E_B_. In this analysis, E_A_ corresponded to one geometrical variable to be tested, and E_B_ was *P_r_* determined numerically. The same 359 data points used for developing the neural network were used for this analysis. The results of this statistical analysis for five potential geometrical factors are presented in Table 2. The results show that *x_p_z_p_/L_x_L_z_* had, in agreement with the results of the previous section, the highest negative PCC with *P_r_*. The *x_p_/L_x_* ranked second because it was equivalent to the square root of the *x_p_z_p_/L_x_L_z_*. The volume fraction ϕ also had a significant correlation factor with *P_r_*. The aspect ratio and the relative thickness also had some impact on *P_r_* but to a lesser degree. It is important to note that the five selected geometrical variables are not all mutually independent.

To better comprehend the effect of these geometrical variables on *P_r_*, one needs to examine the permeation process. Gas molecules entering a membrane will diffuse freely through the polymer matrix in the absence of impermeable nanoparticles. In that case, based on an elementary unit, the entire surface area of the polymer *L_x_L_z_* is available for diffusion. When a nanoparticle is introduced into an elementary unit, the gas molecules have to adopt a more tortuous path to diffuse around the impermeable nanoparticle. Figure 7 presents a plot of the isoconcentration lines within the polymer where the concentration within the cuboid particle is zero. Since the diffusion streamlines of gas molecules in the presence of an impermeable cuboid nanoparticle run perpendicular to the isoconcentration lines, one can easily imagine the diffusion path of these gas molecules. Indeed, the diffusion streamlines are perpendicular to the *x-z* plane of the elementary unit at the gas-membrane interface and deviate progressively as they approach the impermeable nanoparticle. The density of the isoconcentration lines is indicative of the intensity of the local flux. The flux increases when the diffusion channel narrows down and decreases when it widens up.

Considering the dependence of the local permeation flux on the size of the permeation channel for a given *y*-position, a dimensionless parameter *A**, defined in Equation (32), was introduced. This dimensionless parameter was simply the ratio of the projected area that is available for diffusion (*L_x_L_z_ − x_p_z_p_*) and the maximum surface area for diffusion (*L_x_L_z_*). It is logical to postulate that an increase in *A** will lead to an increase in *P_r_* of the MMM, and vice versa. Figure 8 clearly shows the strong relationship between *P_r_* of a MMM with cuboid nanoparticles and the dimensionless parameter *A**.
(32)$$A^{*}=\frac{\left(L_{x}L_{z}-x_{p}z_{p}\right)}{L_{x}L_{z}}$$
where *x_p_z_p_* is the projected area of the nanoparticle, while *L_x_L_z_* is the total permeation area of an elementary membrane unit.

In addition to the influence of *A** on *P_r_*, Figure 8 illustrates the impact of the relative thickness *y_p_/L_y_* over the range spanning from 0.0500 to 0.9833. For a nanoparticle with a fixed *y_p_/L_y_*, a decrease in *x_p_z_p_* leads to an increase of *A** and an increase in *P_r_*. For a fixed *A**, *P_r_* decreases when *y_p_/L_y_* increases. The plot of *P_r_* versus *A** at the largest *y_p_/L_y_* is approaching the 45° line. In other words, *P_r_* approaches *A** when *y_p_/L_y_* tends to 1.0. It is clear that the model to be developed must include the strong linear relationship of *P_r_* with *A** and a nonlinear component to account for the effect of *y_p_/L_y_*.

To characterize and better understand the contribution of the nonlinear portion (NLP) of the relative permeability curve, Figure 9 is presented to highlight the breakdown of *P_r_* in Figure 8 into its linear (*P_r_ = A**) and NLP as a function of *A**. This data exploration is essential in searching for the right form of the analytical equation in the development of an accurate predictive model. The contribution of the NLP that is related to *y_p_/L_y_* is clearly illustrated in Figure 9. The nonlinear term first increases rapidly with A* up to a maximum value before decreasing more gently to zero as *A** tends to unity. The location and the magnitude of the maximum are a function of the relative thickness *y_p_/L_y_*. The maximum value is located in the interval of *A** between 0.16 to 0.38. The contribution of the nonlinear part can be as high as 0.37 of the value of *P_r_* at *y_p_/L_y_* = 0.05 and *A** = 0.18.

Based on the insight of the above information, a new analytical model for the prediction of *P_r_* is now presented in Equations (33) to (35). The model depends strictly on two simple geometrical parameters: *A** and *y_p_/L_y_*. Equation (33) is the sum of the linear and nonlinear parts of the estimated *P_r_*. The parameters *a* and *b* in Equation (33), given by Equations (34) and (35), respectively, are a function of only the relative thickness. It is interesting to note that ϕ is not used explicitly in Equation (33). On the other hand, both *A** and *y_p_/L_y_* indirectly determine the value of ϕ.
(33)$$P_{r}{}_{New}=\frac{aA^{*}}{A^{*}+2a-1}-0.5A^{*}{}^{b}+A^{*}$$
(34)$$a=0.75-0.5\left|0.5-\frac{y_{p}}{L_{y}}\right|$$
(35)$$b=1.5\left(1-\frac{y_{p}}{L_{y}}\right)$$

Figure 10 shows the comparison between the relative permeability obtained by solving Fick’s second law of diffusion numerically and the one predicted with the proposed model. To assess the accuracy of the model prediction, the average prediction error was calculated using Equation (36).
(36)$$\varepsilon_{A}=\sqrt{\frac{1}{n}\sum_{i=1}^{n}{\left(P_{r}{}_{i}-{\hat{P}}_{r}{}_{i}\right)}^{2}}$$
where *ε_A_* is the average prediction error, n is the number of data for the average error calculation, and *P_r_* and ${\hat{P}}_{r}$ are the relative permeability obtained numerically and the one calculated by the proposed model, respectively. Table 3 summarizes the average prediction error for the combination of the geometrical parameters presented in Figure 10. In this investigation, 359 numerically predicted values of the relative permeability were used to fit the model by minimizing Equation (36) using the Levenberg–Marquardt algorithm.

Given its simplicity, the proposed model predicts *P_r_* accurately over a wide range of *A** and *y_p_/L_y_*. Table 3 shows that for each group, the relative particle thickness, *ε_A_* is always below 0.03. The proposed model resorting to two simple geometrical parameters can be used with confidence to predict the permeability of MMMs with impermeable cuboid nanoparticles.

### 4.4. Prediction of Experimental Data with the Proposed Model

To validate the proposed model for the prediction of the relative permeability of MMMs with elongated impermeable nanoparticles, the data of a recent paper published by Zahid et al. [25] was used. This paper reports an investigation on the effect of the morphology of four types of layered graphene flakes, used as fillers, on the gas barrier properties of thermoplastic polyurethane (TPU) films. Fortunately, this paper provides the geometrical parameters of the nanofillers and the experimental permeability data for the oxygen transport across polyurethane films. The digitalized permeability data were converted into relative permeability data using Equation (1). Results in their paper were presented in terms of the mass fraction of the fillers such that, using the bulk density of the four types of layered graphene flake powders provided in their paper and a density of 1.08 g/cm^3^ [26] for the specific thermoplastic polyurethane used, the filler volume fraction was calculated. A void fraction of the nanofiller powder flakes of 0.38 [27] was used in the conversion from mass fraction to volume fraction. The comparison of the experimental relative permeability *P_r_* obtained by Zahid et al. [25] and the *P_r_* predicted by the proposed model is presented in Figure 11. The geometrical parameters *A*^*^ and *y_p_*/*L_y_* depend on the spacing between the particles for fixed particle dimensions; however, this information was not provided by Zahid et al. [25]. Therefore, for each set of experimental results, three values of *A*^*^ were used in the model to obtain reasonable estimations. Correspondingly, *y_p_*/*L_y_* was decreased for an increase of *A*^*^ to maintain constant the filler volume fraction *ϕ*. Results show that the experimental data fall among the predicted *P_r_* curves for the three values of *A*^*^. The proposed model assumed ideal MMMs whereas, in an actual membrane, the dimensions, orientation, and spatial distribution may show significant variability. Despite these potential discrepancies, the comparison of Figure 11 shows that the model can adequately represent the relative permeability of the four types of MMMs presented in the paper of Zahid et al. [25].

On the other hand, there are many studies in the literature where the relative permeability increases with the impermeable filler volume fraction and many studies where the trend varies significantly with the volume fraction [3]. Many reasons were postulated to explain these discrepancies. We strongly believe that the comparison of the relative permeability of the experimental data with the data of ideal MMMs, calculated with the proposed model, could help in finding the sources of the non-ideality. The ideal MMM serves as a benchmark or a point of reference to explain these non-idealities. 

## 5. Discussion and Conclusions

In this paper, the permeation process of gas molecules through MMMs embedding impermeable spherical and cuboid nanoparticles of different aspect ratios was simulated using a three-dimensional finite-difference solution of Fick’s second law of diffusion. These numerical simulations allowed assessing the effect of the shape and size of nanoparticles on the barrier properties of the resulting MMMs. At the same time, these simulations, which can be referred to as numerical experiments, allowed gaging the applicability of different analytical models to predict the relative permeability of these membranes.

Simulation results showed that, for the same volume fraction, the relative permeability of MMMs with cuboid nanoparticles with an aspect ratio larger than unity was lower than the relative permeability of membranes with spherical nanoparticles. The relative permeability decreases with an increase in the aspect ratio of cuboid nanoparticles. The Maxwell’s and similar models predict the relative permeability of MMMs with spherical nanoparticles very well, within a wide (small and medium) particle size range. However, the Maxwell’s model and other analytical models developed for spherical nanoparticles, showed a weak prediction of the relative permeability of MMMs with cuboid nanoparticles. Using an artificial neural network, we developed a feedforward neural network model, which very accurately predicts the relative permeability of MMMs with both spherical and cuboid nanoparticles for a wide range of sizes and the aspect ratios. However, although very accurate, the developed feedforward neural network model lacks analytical models’ simplicity and explicability. To overcome this limitation of neural network-based models, we used the Pearson correlation coefficient and performed a multivariate covariance analysis to elucidate the relationship between the shape and the relative dimensions of nanoparticles and the relative permeability of MMMs. This led to an analytical model, which involves only two geometrical parameters, the ratio of the projected area available for diffusion and the maximum surface area for diffusion, which depends on the aspect ratio of the nanoparticle, and the relative thickness of the nanoparticle. Despite its simplicity, the new model accurately predicts the relative permeability of the MMMs with cuboid nanoparticles for a wide range of sizes and aspect ratios. To our best knowledge, the model developed in this study is the first one in the literature, which is applicable for MMMs with both spherical and layered nanoparticles.

The paper assumed an ideal case where the MMMs had ideal interfacial compatibility, no particle agglomeration, and particles were perfectly aligned. However, in reality, voids could exist due to poor adhesion between fillers and the polymer matrix; agglomeration of particles may create nanogaps between the particles and the polymer chains [9]. These non-ideal morphologies may increase or decrease the *Pr* such that the effects of non-ideality need to be further investigated even though it is very challenging to represent non-ideality in practice. One needs to idealize non-ideal morphology for modelling. This investigation on ideal MMMs is the first, but very important, step to develop an accurate model for MMMs with cuboid nanoparticles. Indeed, as noted by Vinh–Thang and Kaliaguine [9], good predictions are only achieved for MMMs in which morphology could be represented as a combination of voids with an ideal filler/polymer interface. In addition, as suggested by Hamid and Jeong [28,29], the embedding of nanosized particles within the polymer matrix will favor a uniform distribution of the nanoparticles and better wetting property between the filler phase and the polymer phase [30] compared to the distribution of micron-sized particles, which are often used in the fabrication of MMMs. The smaller size particles will allow the fabrication of thin membranes and will enhance filler-polymer interface contact.

## Figures and Tables

**Figure 1 membranes-10-00422-f001:**
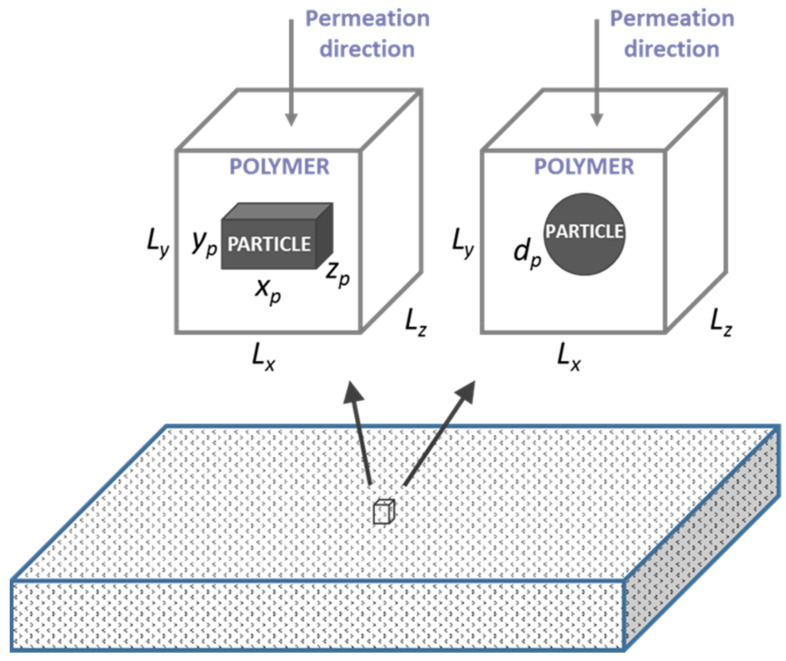
Schematic diagram of a MMM and one elementary unit with a nanoparticle located at its center; *y* is the direction of the gas permeation. The dimensions of the cuboid nanoparticle are *x_p_*, *y_p_*, and *z_p_*, and the spherical particle is *d_p_*. The dimensions of the polymer elementary unit are *L_x_*, *L_y_*, and *L_z_*.

**Figure 2 membranes-10-00422-f002:**
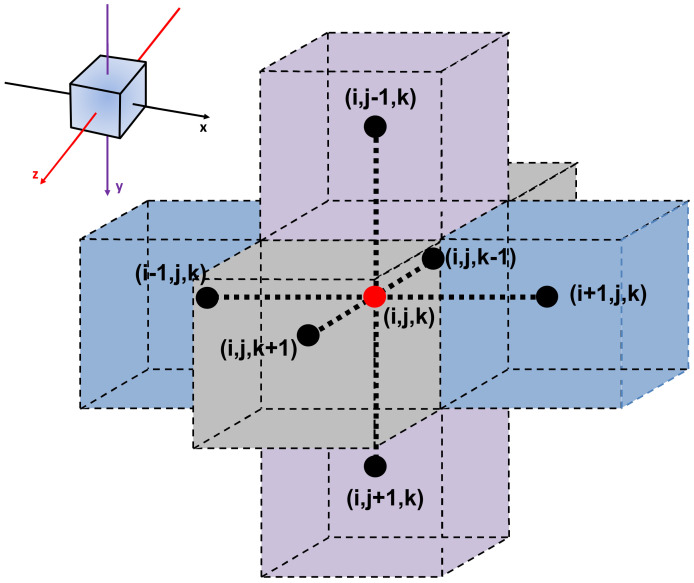
Nomenclature of an interior mesh point with its six neighboring mesh points. *x*, *y*, and *z* directions are represented by *i*, *j*, and *k*, respectively.

**Figure 3 membranes-10-00422-f003:**
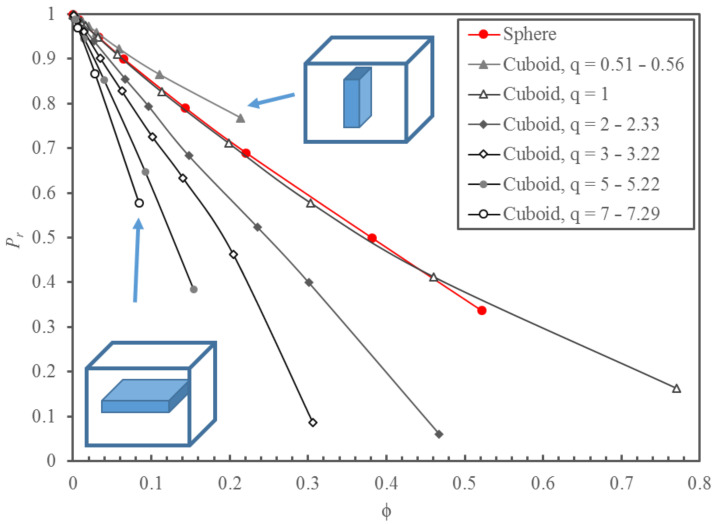
Plots of the relative permeability *P_r_* of MMMs as a function of the solid volume fraction ϕ for spherical and cuboid impermeable nanoparticles, grouped by narrow ranges of the aspect ratio *q*. Data points were obtained numerically, and the solid lines are trend lines. The permeability of the continuous phase was kept constant at 5 × 10^−12^ m^2^/s (5.0 × 10^−11^ m^2^/s diffusivity and 0.10 solubility).

**Figure 4 membranes-10-00422-f004:**
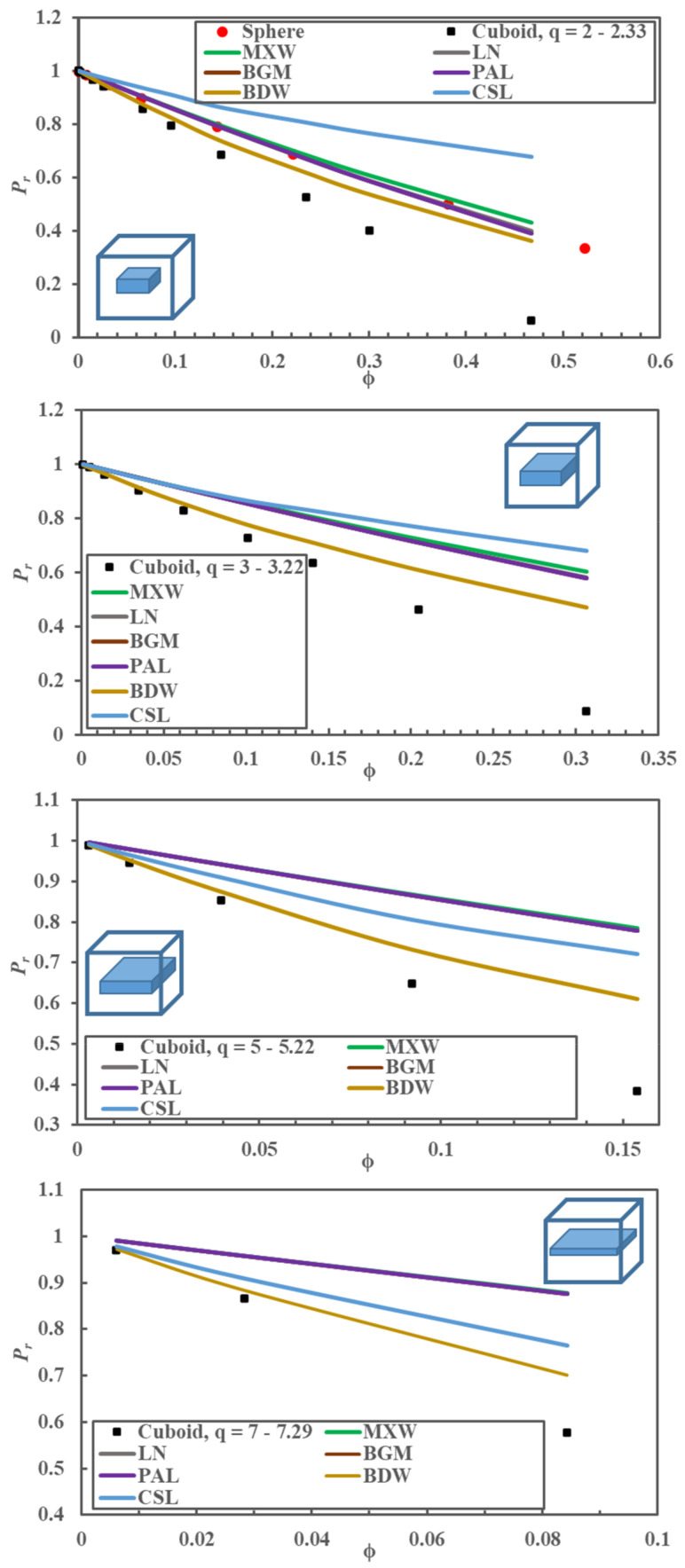
A comparison of the relative permeability of MMMs obtained numerically with the ones predicted by the six models presented in Table 1 as a function of the filler volume fraction (ϕ). The comparison is made for a polymeric elementary unit containing an impermeable nanoparticle: A sphere and cuboids with four different ranges of the aspect ratio (*q*) ([2.0, 2.3], [3.0, 3.2], [5.0, 5.2], [7.0, 7.3]). Acronyms of each model are defined in Table 1.

**Figure 5 membranes-10-00422-f005:**
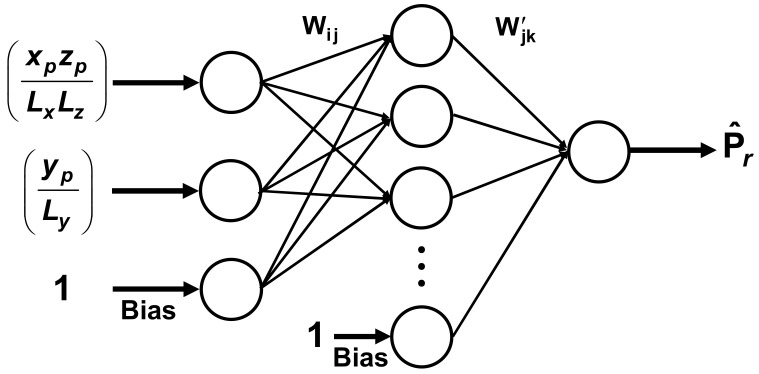
Feedforward neural network for the prediction of the relative permeability (*P_r_*) for impermeable cuboid nanoparticles as a function of the normalized projected area (*x_p_z_p_*/*L_x_L_z_*) and the relative thickness (*y_p_*/*L_y_*).

**Figure 6 membranes-10-00422-f006:**
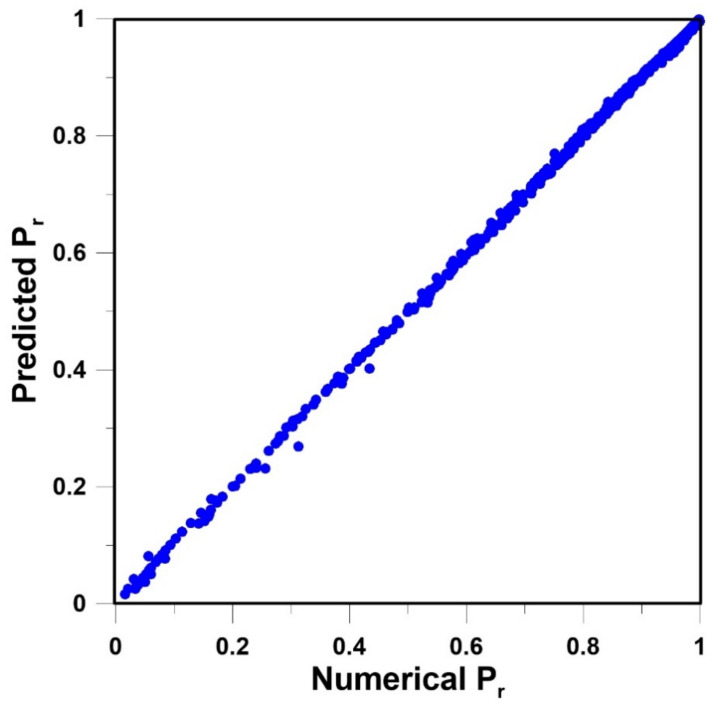
Parity plot of the predicted *P_r_* and the numerically-determined *P_r_* for the feedforward neural network (FFNN) of Figure 5 with six hidden neurons, including the bias.

**Figure 7 membranes-10-00422-f007:**
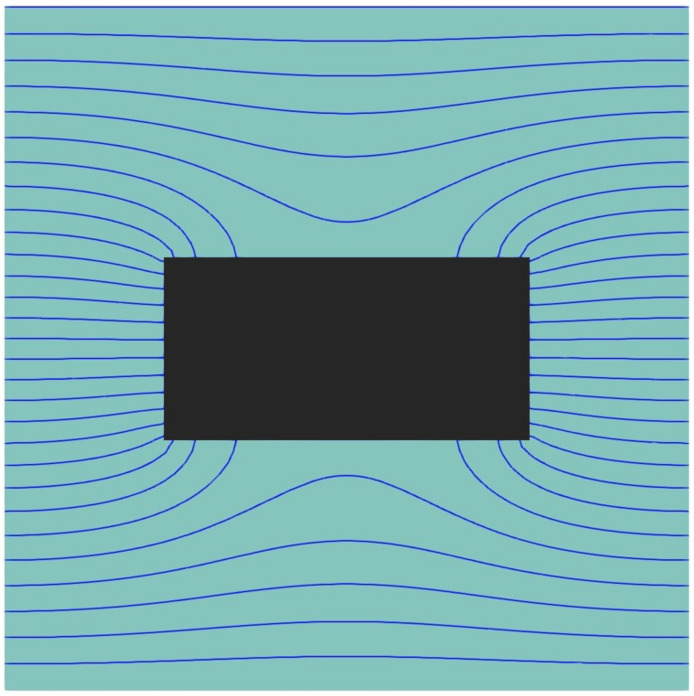
The steady-state isoconcentration lines of the migrating species on the half-cut plane of an elementary unit of a MMM with a cuboid nanoparticle with an aspect ratio *q* = 2.07 and a relative thickness *y_p_/L_y_* = 0.25.

**Figure 8 membranes-10-00422-f008:**
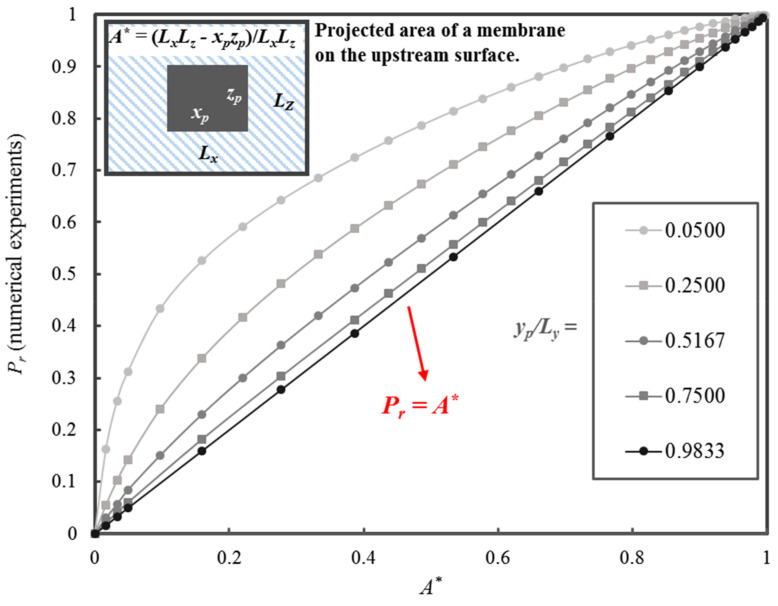
*P_r_* of MMMs as a function of *A** for five values of *y_p_/L_y_* ranging from 0.0500 to 0.9833.

**Figure 9 membranes-10-00422-f009:**
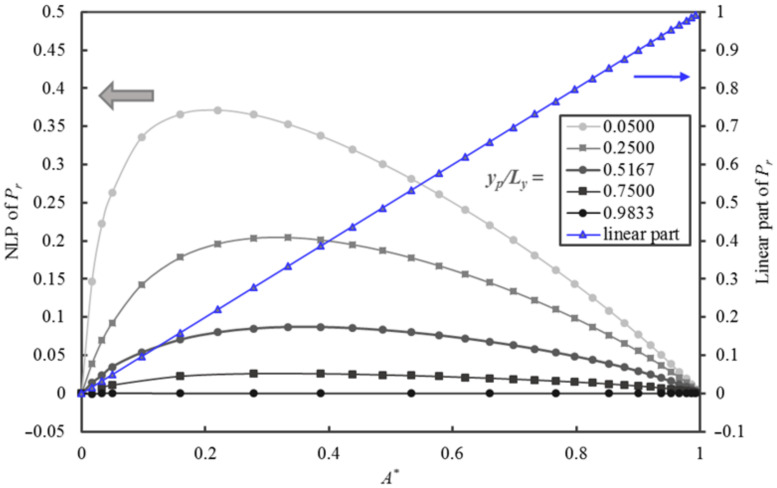
Plots of the linear and nonlinear portions of the *P_r_* of a MMM as a function of *A** for five values of *y_p_/L_y_* ranging from 0.05 to 0.98.

**Figure 10 membranes-10-00422-f010:**
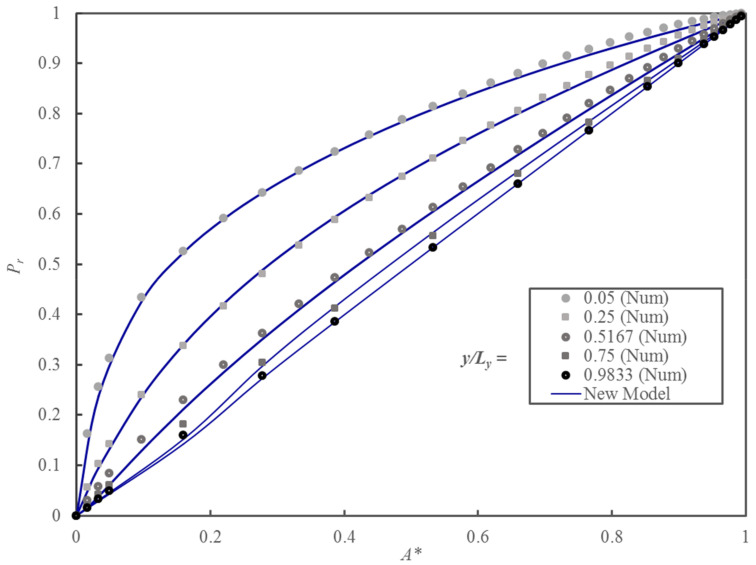
Plots of the *P_r_* of MMMs embedding cuboid nanoparticles as a function of *A** for five different values of *y_p_/L_y_*.

**Figure 11 membranes-10-00422-f011:**
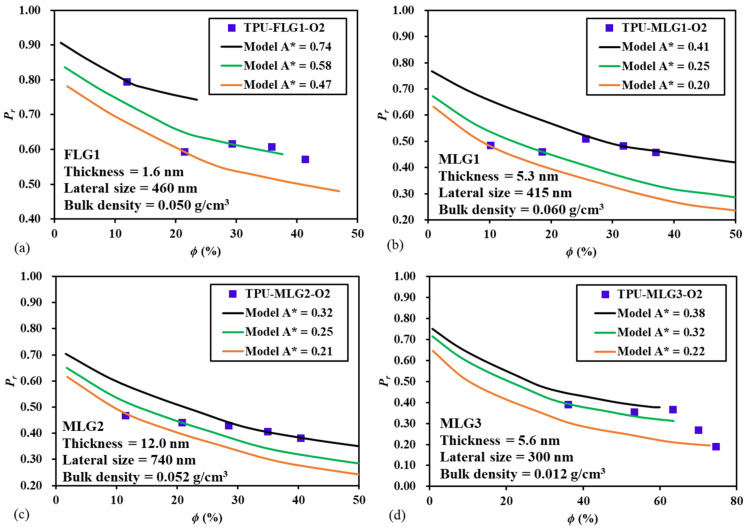
Comparison of the experimental data of Zahid et al. [25] and the prediction of the proposed model for the relative permeability of the oxygen transport across four types of MMMs consisting of thermoplastic polyurethane (TPU), embedding few-layer (FLG), and multi-layer (MLG) graphene flakes.

**Table 1 membranes-10-00422-t001:** Predictive models for the relative permeability (*P_r_*) of a migrating species in a mixed-matrix membrane (MMM) with nanoparticles [19].

Model	Equation	Equation #
Maxwell (MXW) [10]	$P_{r}=\frac{P_{d}+2P_{c}-2\phi (P_{c}-P_{d})}{P_{d}+2P_{c}+\phi (P_{c}-P_{d})}$	(2)
	$\begin{array}{cc} P_{d}=0: & P_{r}=\frac{1-\phi }{1+\phi /2} \end{array}$	(3)
Bruggeman (BGM) [11]	$P_{r}{}^{1/3}\left(\frac{P_{d}-P_{c}}{P_{d}-P_{r}P_{c}}\right)={\left(1-\phi \right)}^{-1}$	(4)
	$\begin{array}{cc} P_{d}=0: & P_{r}={\left(1-\phi \right)}^{3/2} \end{array}$	(5)
Lewis–Nielsen (LN) [12,13]	$P_{r}=\frac{1+2\phi \left(P_{d}-P_{c}\right)/\left(P_{d}+2P_{c}\right)}{1-\psi \phi \left(P_{d}-P_{c}\right)/\left(P_{d}+2P_{c}\right)},\;\psi =1+\left(\frac{1-\phi_{m}}{\phi_{m}{}^{2}}\right)\phi$	(6)
	$\begin{array}{cc} P_{d}=0: & P_{r}=\frac{1-4\phi }{1+2\psi \phi } \end{array}$	(7)
Pal (PAL) [14]	$P_{r}{}^{1/3}\left(\frac{P_{d}-P_{c}}{P_{d}-P_{r}P_{c}}\right)={\left(1-\frac{\phi }{\phi_{m}}\right)}^{-\phi_{m}}$	(8)
	$\begin{array}{cc} P_{d}=0: & P_{r}={\left(1-\frac{\phi }{\phi_{m}}\right)}^{3/2\phi_{m}} \end{array}$	(9)
Cussler (CSL) [15,16]	$\begin{array}{cc} P_{r}=\frac{1}{1+\alpha \phi }, & \alpha \phi <1 \end{array}$	(10)
	$\begin{array}{cc} P_{r}=\frac{1-\phi }{1-\phi +\mu \alpha^{2}\phi^{2}}, & \alpha \phi >1 \end{array}$ *µ* = 1 for flakes as periodic ribbons, *µ* = 4/9 for flakes as periodic hexagons.	(11)
Bharadwaj (BDW) [17]	$P_{r}=\frac{1-\phi }{1+\frac{x_{p}}{3y_{p}}\left(\mu +\frac{1}{2}\right)\phi }$, *µ* = 0 for randomly dispersed fillers, *µ* = 1 for fillers perfectly aligned perpendicular to the gas flux	(12)

**Table 2 membranes-10-00422-t002:** Multivariate covariance analysis of geometrical factors and *P_r_*.

Pearson Correlation Coefficient (PCC)					
Rank	Variable	Average	*σ* _EA_	Cov(E_A_, *P_r_*)	PCC(E_A_, *P_r_*)
1	*x_p_z_p_/L_x_L_z_*	0.3381	0.3154	−0.0865	−0.9629
2	*x_p_/L_x_*	0.4961	0.2923	−0.0763	−0.9156
3	ϕ	0.1331	0.1821	−0.0436	−0.8401
4	*q*	2.5620	3.4326	−0.1937	−0.1981
5	*y_p_/L_y_*	0.4022	0.2753	−0.0144	−0.1831
-	*P_r_*	0.7395	0.2850	-	-

**Table 3 membranes-10-00422-t003:** Average prediction errors of the *P_r_* of Figure 10 using the proposed model evaluated over five values of the *y_p_/L_y_*.

*y_p_/L_y_*	*A**	*ε_A_*
0.0500	0.03–0.99	0.0098
0.2500	0.03–0.99	0.0070
0.5167	0.03–0.99	0.0110
0.7500	0.16–0.99	0.0281
0.9833	0.16–0.99	0.0048

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
