# Peer review of "Gas Permeation Model of Mixed-Matrix Membranes with Embedded Impermeable Cuboid Nanoparticles"

_membranes, 2020, doi:10.3390/membranes10120422_

Round 1

Reviewer 1 Report

Recommendation: Publish after major and minor revisions noted.

This work seems to be quite interesting and relevant. The authors proposed a new model for the calculation of gas relative permeability (ratio of membrane permeability to the permeability into the pure polymeric matrix) in mixed matrix membranes (MMM) formed by cuboidal nanoparticles over an extensive range of the solid volume fraction and aspect ratio. Such a model considers the MMM as isotropic media, that is, the impermeable particles have a monodisperse size distribution, the same shape and would be uniformly distributed through the polymeric matrix. In addition, MMM would not present extra free volume, due to the low adhesion along the solid/polymer interfaces. Although simple and limited, the model represents some real physical situations. This article presents an elegant approach and produces a really promising result; however, its contribution still needs to be confirmed. Therefore, my recommendation is to publish in MEMBRANES after some major and minor revisions.

The work developed in this article could be divided into four stages: (i) generation of gas relative permeability data into MMM with cuboid and spherical particles over a wide range of solid compositions and geometrical parameters. Such data were calculated from three-dimensional numerical simulations carried out to describe the mass transfer of gases in films in the timelag experiments; (ii) evaluation of the predictive capability of some analytical models available in the literature based on the data generated in this work; (iii) fit of an artificial neural network to calculate these data; and (iv) the proposal of a new analytical model for the prediction of the relative permeability of gases in isotropic MMM formed by impermeable cuboid particles as a function of solid volumetric composition and aspect ratio. In this last step, the authors developed a multivariate covariance analysis to identify the most appropriate functional form to relate the permeability relative to the composition and the particles geometry.

I remark some major comments:

  • As previously described, the authors used a single MMM model to develop their analyzes and proposals. The relevance of this work will only become clear if, initially, the authors evaluated, which kind of real systems could be adequately described by this model. Table 1 of the Wolf et al. (2018) paper presents several sources of experimental data that could be used here. In this sense, I strongly suggest that the authors develop a critical analysis, identifying, among the membranes experimentally studied in literature, those that could adequately be described by the isotropic solid model adopted in this work. For such membranes, examples should be presented in supplementary material or, if any, published works with them should be cited, confirming the fitting of the three-dimensional transport model proposed in this work to the experimental curves obtained by timelag. Thus, the authors not only highlight the relevance of the work, but also exemplify how the experimental characterization of MMM could be used to obtain the parameters of the proposed model.
  • At the end of the work, the proposed analytical model must be tested against the experimental data highlighted at the previous item;
  • The review of the analytical predictive models was limited to general aspects, without explaining possible theoretical bases, the parameter ranges and the expected errors for MMM with different morphologies. Please, complete your review with more details.
  • Would the trends observed in Figure 3 be the same for real membranes formed with the same solid volume fractions and aspect ratios, but involving different polymers? Even in MMM with good solid / polymer adhesion, it will be expected that the chemical and physical properties of the polymer influence the behavior of relative permeability. Please clarify with real examples?
  • Why were the simulations limited up to different solid compositions when change the aspect ratio (q)?
  • In Figure 4a (explanation in lines 237-241), why does Maxwell's model deviate at high solid compositions (stating in the vicinity of 0.25), while BGM, LN and PAL show good results even at higher particle contents? This type of doubt could be avoided in view of the necessary review in the comment (3).
  • Which was the database, the objective function and the minimization method for obtaining the parameters of the Eqs. (27-29);
  • 5. Conclusion section should be improved to better highlight your contributions and should be more concise. The authors repeat arguments already used in the introduction and conclude on points not explored in the work.

I also remark some minor comments:

  • Line 77 - Please align first line to the left margin;
  • Line 195 – Please, correct subscript 0 and Ly at the Jy symbol. Take the opportunity to check the symbology adopted throughout the manuscript;
  • Lines 229-231 – The caption in Figure 3 should present the values of sorption and diffusion coefficients (or permeation coefficient in the pristine polymer) used into simulations. Please, also clarify if these parameters were kept constant during simulations;
  • Figure 4 should be improved, Improving the color contrast between lines;
  • Line 325 - It is necessary a reference for the covariance analysis method using Pearson correlation coefficient (PCC). In this analysis, which data were used: those obtained from the simulations or the ones generated by the artificial neural network? Please explain;
  • Lines 327-329 – Please, maintain consistency between the symbols displayed in Eqs. (23-25) and those presented throughout the text;
  • Please, correct the “diffusion path pf these gas” for “diffusion path of these gas”;
  • Finally, the reference list must be revised to attend the Journal rules.

Reviewer 2 Report

The work "Gas permeation model of mixed-matrix membranes
with embedded impermeable cuboid nanoparticles", regards an important field of application for the realization of advanced devices based on MMMs. So the theoretical rationalization of wich mathematical model is the most appropriate, discussed in the introduction is of high interest for the generic study of any permeation phenomenon. Morover, the computational study conducted and the data presentation is clear and usuful for the scientific divulgation and advancement of knowledge for users that daily move their endeavour to obtain new defect-free membranes. Concluding, I aknowledge authors for the choice of the subject and I appreciate the work well done here presented. 

Round 2

Reviewer 1 Report

Recommendation: Publish after minor revision noted.

In this version of the manuscript, the authors answer the main remarks highlighted at the first version. Therefore, my recommendation is to publish in MEMBRANES after one minor revisions:

I agree with the authors when they postulate that: "As clearly discussed by Wolf et al. [3] and in numerous other papers, there is a wide variability in the results of the relative permeability. It is not rare to find in the literature data where the permeability of MMMs with impermeable fillers increases with the mass or volume fraction. In some studies, the relative permeability decreases very rapidly with the filler loading whereas, in other instances, the relative permeability increases and then decreases or vice versa. All these authors postulate hypothesis attempting to explain the behaviors that are observed. With the tremendous recent growth in the development of MMMs, it is important to understand the reasons of the trends observed. To assist in finding the reasons why the observed relative permeability does not follow the expected trend, it is important to have a point of comparison or a benchmark corresponding to an ideal MMM. This what the proposed model, as a point of reference, can provide." It is really a relevant and helpful contribution to this scientific area. Therefore, make it clear at the work objectives, even if it has already presented in the discussion section (§4.4).
